# Modeling of the chest wall response to prolonged bracing in pectus carinatum

Brandon Sargent[1]*, Katie Varela[2], Dennis Eggett[3], Emily McKenna[4], Christina Bates[4], Rebeccah Brown[4], Victor Garcia[4], Larry Howell[2]

**1** Department of Mechanical Engineering, Gonzaga University, Spokane, Washington, United States of America, **2** Department of Mechanical Engineering, Brigham Young University, Provo, Utah, United States of America, **3** Department of Statistics, Brigham Young University, Provo, Utah, United States of America, **4** Chest Wall Center, Cincinnati Children's Hospital Medical Center, Cincinnati, Ohio, United States of America

* sargentb@gonzaga.edu

**Data Availability Statement:** The data has been uploaded to our institution's open access archive, BYU Scholar's Archive: https://scholarsarchive.byu.edu/data/53.

## Abstract

Pectus carinatum is a chest wall deformity that is often treated through the wearing of an external brace. The treatment of the deformity could benefit from a greater understanding of chest wall characteristics under prolonged loading. These characteristics are difficult to model directly but empirical studies can be used to create statistical models. 185 patients from 2018-2020 received bracing treatment. Data on the severity of the deformity, treatment pressures, and time of wear were recorded at the first fitting and all subsequent follow-up visits. This data was analyzed using a statistical mixed effects model to identify significant measures and trends in treatment. These models were designed to help quantify changes in chest wall characteristics through prolonged bracing. Two statistical models were created. The first model predicts the change in the amount of pressure to correct the deformity after bracing for a given time and pressure. The second model predicts the change in pressure response by the body on the brace after bracing for a given time and pressure. These models show a high significance in the amount of pressure and time to the changes in the chest wall response. Initial deformity severity is also significant in changes to the deformity. The statistical models predict general trends in pectus carinatum brace treatment and can assist in creating treatment plans, motivating patient compliance, and can inform the design of future treatment systems.

## Introduction

Pectus carinatum (PC) is a chest wall deformity in which part of the chest wall protrudes outward. It can cause chest pain and discomfort, changes to cardiac and respiratory efficiency, as well as significant psychological impact, especially in adolescents [1–3]. Currently, external braces constitute the majority of treatment methods [4, 5] but in some cases, surgery is performed [6–9]. The FMF Dynamic Compressor System, developed in 2001, is an example of a brace used to correct PC and is widely used today [3].

When designing treatment methods for PC and other chest wall deformities (such as pectus excavatum), a clear understanding of the physical response under prolonged loading of the

**Funding:** Funded Study: BS, VH, VG, and LH received a funded research contract via payments to their respective institutions from Zimmer Biomet. The funding institution website is zimmerbiomet.com. NO, the funders had no role in study design, data collection and analysis, decision to publish, or preparation of the manuscript.

**Competing interests:** The authors have declared that no competing interests exist.

chest wall is crucial in developing a proper treatment method [10–14]. However, due to the growth of adolescents, human variability, the interactions of different tissues, and the ability of the body to remodel, physical characteristic of large anatomical segments such as the chest wall can be difficult to model [11, 15–18]. A complete model would be required to include tissue interactions, bone density and health, in addition to the properties of cartilage, muscles, skin, etc. all of which can remodel under prolonged loading. Simplified models that combine these remodeling characteristics into simple parameters, while less individually precise, can provide benefits to medical device manufactures and clinicians in developing treatments that can address a large patient population [19–21].

Specific to PC, understanding the chest wall response to bracing may be useful in proscribing the pressure the brace should exert and the amount of time it should be worn [3]. As the chest wall begins to remodel towards the corrected position, treatment times and pressures could change [4]. Being able to characterize or predict some of that changing behavior in patients given initial tests and prescribed treatments could enable better treatment options for patients. This could help motivate patient compliance, a large concern when it comes to bracing [10, 22].

Pectus carinatum is a chest wall deformity, specific types of which have been nicknamed "pigeon breast" and "chicken breast", in which the wall protrudes outward. It is more common in males than females [23] at approximately a 4:1 ratio [11, 24]. PC consists of about 15% of chest wall deformities [11] with an overall prevalence of 0.6% and, while the etiology is unclear, it may be in part influenced by other medical conditions [23, 25, 26]. Perhaps because Carinatum is less intrusive into the body than Pectus Excavatum, there are significantly fewer surgeries performed on individuals who suffer from this deformity [6]. Bracing has become a more common treatment for the individuals with PC [3, 4, 23]. The brace used in this study, the FMF Dynamic Compressor System, is adjustable and is fitted to each patient; made from curved aluminum segments, shoulder straps, cushioned compression plates, and pads to cushion the defect as shown in Fig 1. An instrumented measuring pad is also used to assist in fitting [27, 28].

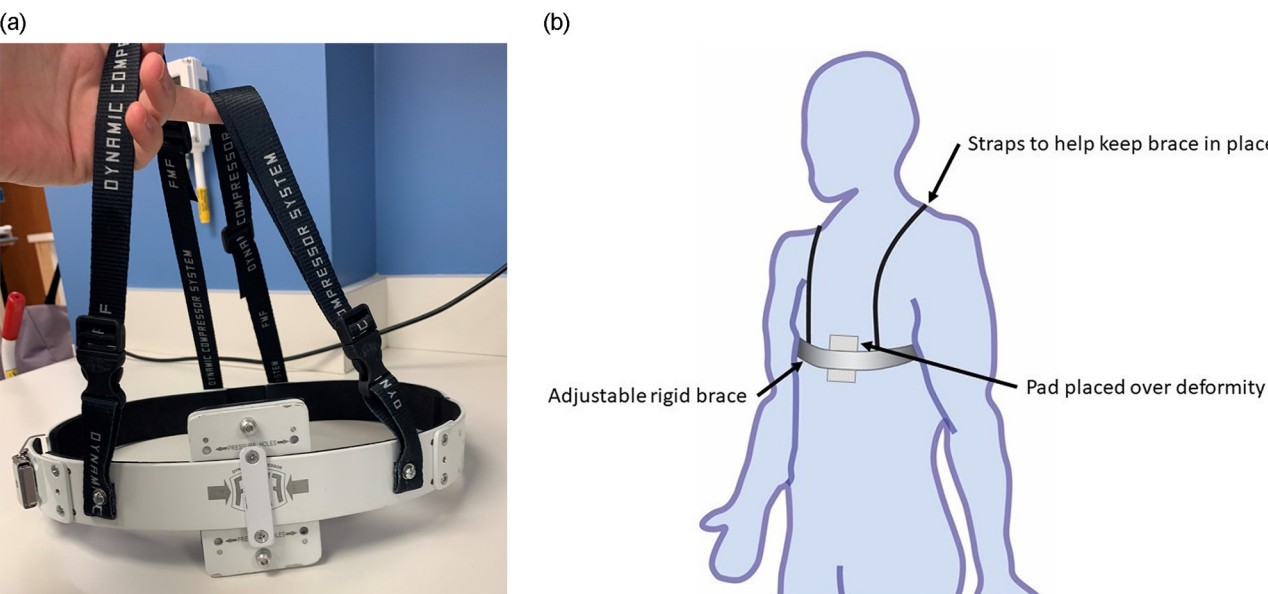

(a)

(b)

Straps to help keep brace in place

Adjustable rigid brace

Pad placed over deformity

**Fig 1. The brace system used in this study.** The system is made from curved aluminum segments, shoulder straps, cushioned compression plates, and pads to cushion the defect. The clinical also uses an instrumented pad to obtain measurements and assist in fitting the brace.

During the initial visit and each subsequent follow-up visit, the clinical staff and orthotist, referred to here collectively as the clinicians, use an instrumented pad to push the chest extrusion to the corrected position. This pad would read how much pressure (measured in pounds per square inch, PSI) was required to perform this action, known as the pressure of correction, or *POC*. This is a one-time value representative of the instantaneous state of the deformity. An understanding of this value has been used to guide treatment methods [8]. As the brace is worn and the deformation slowly corrected, this value is expected to decrease with time.

A typically smaller pressure, *PT*, is the prescribed pressure for treatment. This is a value determined by the clinician. The brace is then fitted such that it exerts this amount of pressure on the patient at the time it is prescribed.

As the patient's anatomy begins to remodel, the deformity does not resist the brace as much as when it was fitted. The brace, fixed at a constant fitting configuration, does not change as the body remodels. Therefore the pressure at the deformity/brace interaction point changes with time of wearing the brace at a prescribed *PT*. This new pressure "prior PT", measured at the subsequent follow-up visit, is the pressure the body is feeling now at the brace fitting configuration set at the prior visit. The difference between the *PT* prescribed at the previous clinical visit and the prior PT value measured at the current clinical visit represent the change in pressure exerted by the body on the pad over the time between visits.

This work analyzes a patient group with PC treated using bracing. A statistical analysis of the patient chest wall response to bracing provides an understanding of general trends in human response and can inform future treatment procedures and products. Models to predict the changes in *POC* and *PT* under prescribed treatments are developed.

## Materials and methods

On January 4, 2021, the Cincinnati Children's Hospital Institutional Review Board (IRB) reviewed this submission. Consent was obtain via written approval. The following statement is a direct quote from the approval document: "The research activities described in the above submission have been determined to be EXEMPT from IRB review in accordance with applicable regulations and institutional policy." From Feburary 2018 to March 2020, the clinical staff at Cincinnati Children's Hospital Medical Center (CCHMC) Chest Wall Center recorded treatment data for 185 PC patients. At the beginning of treatment, the initial POC was measured and a clinical staff member prescribed a PT value and fitted a brace for the patient to provide that pressure. The clinician also recommended a number of hours per day that the patient should wear the brace from 3-6 hours a day to as many hours as possible. The recommended hours of wear was determined by the initial *POC*. Higher values of *POC* were prescribed lower wear times initially to help ease the patient into the treatment plan. This was determined in part due to the large difference between the *POC* and the *PT*. Those whose initial *POC* closer matched the *PT*, the patients could wear the brace more comfortably all day, with the exception of approved activities such as showering or bathing. This was determined by the clinicians, taking into consideration the desire to have as much corrective brace wearing time while promoting patient wearing compliance and comfort. The brace was used with a standard pad measuring 7 cm x 10 cm. The clinicians attempted to center the deformity in the middle of the pad when fitting the brace.

The patient returned periodically for follow-up visits to adjust the brace. A total of 974 follow-up visits were recorded across the 185 patients. At each check-up visits, the prior PT was recorded, as well as the new POC measure. If no pressure was needed to obtain a desirable deformity depth (i.e. the chest naturally rests at a corrected position) then zero is recorded as the value for POC. The patient self-reported the number of hours they wore the brace daily.

While patient-reported wear carries with it inherent difficulties due to accuracy, but to avoid the use of powered electronics on the brace throughout the entire wear time, this measure is the best available measure.

Depending on the new POC value and the prior PT value and the patient compliance, the clinician adjusted the brace to exert a new PT value. This was repeated until the clinician determined that treatment was complete and the brace was no longer needed to maintain a corrected chest position or the patient no longer returned for follow-up visits.

A statistical analysis was performed on de-identified data using a mixed effects models procedure to investigate potential trends in the recorded data. This method evaluate visit-to-visit changes and trends for a given subject. This was performed in SAS 94. Each patient's treatment data was used as a subject in the mixed model. The subject models are then evaluated collectively to present the most significant statistical model for the entire group of subjects. Initial *POC*, subsequent *POC* values, *PT*, prior *PT* values, and brace time usage were considered as possible contributions. Each treatment period, or time between clinical visits, was used as an observation. A Type 3 Test of Fixed Effects was also completed to test the significance of each of the fixed effects.

Two primary models were developed. Both models use a pressure-time term, given as

$$P_{mon} = \frac{hrs \times days \times PT}{24 \times 30} \qquad (1)$$

where *hrs* represents the amount of self-reported hours a day the patient wore the brace, *days* represent the days at a given prescribed pressure of treatment, and *PT* is the prescribed pressure of treatment in PSI. This term was used to group the time at a given prescribed pressure into a single variable as both time and the prescribed pressure at that time are interdependent. Instances where the patient reported an hour value that covered a span of days (e.g. 6 hours every other day), the time was divided by number of days in the span and that value was reported as the hours a day for that patient over that period (e.g. 6 hours every other day would be recorded as 3 hours daily). In instances where the patient recorded a range of times for a given day, the upper limit was used (e.g. 6-12 hours a day was recorded as 12 hrs).

## Results

Of 185 patients, all had at least one follow-up visit. The average number of days between follow-up visits was 50 (ranging from 10 to 543, SD: 36.3). No first follow-up visit was less than 18 days from the initial fitting with the average time between the initial visit and first follow-up being 48 days (SD: 45.9). Of the 185 patients, as of February 2021, 50 patients had completed treatment. The average time to treatment completion was 209.5 days (ranging from 28 to 765, SD: 158.8). The average initial *POC* value recorded at the first fitting was 3.8 PSI (range from 1 to 7, SD: 1.4). 84 had not come in for another treatment for more than a year and could be considered as lost to follow-up.

The first of the statistical models developed predicts the *POC* value based on the initial *POC* and a prescribed treatment pressure and time. This model can be used to predict how much pressure and time is needed to correct the deformity, resulting in a *POC* of zero. Data from 182 of the subjects were used in this model with 780 observations. The maximum number of observations for a single participant was 17. The resulting model is

$$POC = 0.7815POC_{Initial} - 0.1714P_{mon} - 0.3011 \qquad (2)$$

where *POC* represents the pressure to correct at a given time, given an initial *POC* value and a

time pressure term that takes into account the treatment pressure, the hours worn a day, and the days worn.

The statistical significance for the effects in Eq 2 for fixed effects are listed in Table 1. The results of the Type 3 Tests for Fixed Effects can be found in Table 1. It can be noted that the intercept shown in Eq 2 was not found to be statistically significant (assuming a threshold of significance of $\alpha = 0.05$). This follows theory, as a zero value of $POC_{initial}$ and $PT_{mon}$ should not result in any change in $POC$. For this reason and the low statistical significance, the model was rerun forcing a zero value for the intercept. That resulted in an updated model of

$$POC = 0.7619POC_{Initial} - 0.2378P_{mon} \tag{3}$$

The statistical significance for the effects in Eq 3 for fixed effects are listed in Table 1. The results of the Type 3 Tests for Fixed Effects can be found in Table 1. It can be noted that both parameters remain highly significant.

Both models were analyzed with respect to the clinical data by comparing the model predicted value for each follow-up visit and the clinical recorded data. The model in Eq 3 proved on average to better predict over the models with a variable intercept (Eq 2). The zero-intercept model in Eq 3 is the $POC$ model used hereafter.

The average difference between the $POC$ recorded value and the model prediction over the 974 follow-up visit observations is 0.53 PSI with a standard deviation of 1.05. This indicates that the model will, on average, predict a greater reduction in POC than what was observed clinically by 0.53±1.05 PSI. Four randomly selected example cases from the data set were selected to validate the model and compare the model predictions to clinical data. For each visit, the model was used to predict how the deformity will change with time given the prescribed treatment plan using the patient reported time of wear. Then at the follow-up visit measurement, the model is updated with the measured $POC$ value. The difference between the model predicted value and the measured value at each follow-up visit was measured. In all the selected cases, each measured valued was within the average difference and standard deviation.

**Table 1. Model effects for POC models (Eqs 2 and 3).**

| Statistical Measures for Fixed Effects Model with variable intercept | | | | | |
|---|---|---|---|---|---|
| Effect | Estimate | Standard Error | DF | t Value | Prob. > \|t\| |
| Intercept | -0.3011 | 0.2433 | 179 | -1.24 | 0.2179 |
| Initial POC | 0.7815 | 0.0508 | 454 | 15.39 | <0.001 |
| Pressure-Months | -0.1714 | 0.0166 | 144 | -10.29 | <0.001 |
| Type 3 Tests for Fixed Effects with variable intercept | | | | | |
| Effect | Numerator DF | Denominator DF | | F Value | Prob. > F |
| Initial POC | 1 | 454 | | 236.77 | <0.001 |
| Pressure-Months | 1 | 144 | | 105.92 | <0.001 |
| Statistical Measures for Fixed Effects Model with fixed zero intercept | | | | | |
| Effect | Estimate | Standard Error | DF | t Value | Prob. > \|t\| |
| Initial POC | 0.7619 | 0.0124 | 600 | 61.55 | <0.001 |
| Pressure-Months | -0.1714 | 0.0166 | 178 | -9.02 | <0.001 |
| Type 3 Tests for Fixed Effects with fixed zero intercept | | | | | |
| Effect | Numerator DF | Denominator DF | | F Value | Prob. > F |
| Initial POC | 1 | 600 | | 3788.95 | <0.001 |
| Pressure-Months | 1 | 178 | | 81.39 | <0.001 |

**Table 2. Model effects for ΔPT models (Eqs 4 and 5).**

| Statistical Measures for Fixed Effects Model with variable intercept | | | | | |
|---|---|---|---|---|---|
| Effect | Estimate | Standard Error | DF | t Value | Prob. > \|t\| |
| Intercept | -0.2105 | 0.1425 | 181 | -1.48 | 0.1414 |
| Pressure-Months | 0.2885 | 0.0750 | 144 | 3.85 | <0.001 |
| Type 3 Tests for Fixed Effects with variable intercept | | | | | |
| Effect | Numerator DF | Denominator DF | | F Value | Prob. > F |
| Pressure-Months | 1 | 144 | | 14.80 | <0.001 |
| Statistical Measures for Fixed Effects Model with fixed zero intercept | | | | | |
| Effect | Estimate | Standard Error | DF | t Value | Prob. > \|t\| |
| Pressure-Months | 0.1975 | 0.0481 | 178 | 4.11 | <0.001 |
| Type 3 Tests for Fixed Effects with fixed zero intercept | | | | | |
| Effect | Numerator DF | Denominator DF | | F Value | Prob. > F |
| Pressure-Months | 1 | 178 | | 16.86 | <0.001 |

A second model represents the change in treatment pressure between follow-up dates and can be used to predict prior *PT* values. This will represent the decrease in the pressure exerted by the brace on the deformity over a treatment period. Data recorded from 184 of the subjects were usable in this model with 783 observations total. The maximum number of observations from a single subject was 17.

$$\Delta PT = 0.2885 P_{mon} - 0.2105 \quad\quad\quad (4)$$

This equation can be used to determine the change in treatment pressure between the body and the brace during a treatment period. For example, if the initial prescribed pressure is a *PT* of 1.5 PSI and on the subsequent follow-up visit, the prior PT value is measured as 1 PSI, the ΔPT value for that interval would be 0.5 PSI. The statistical significance for the effects in Eq 4 for fixed effects are listed in Table 2. The results of the Type 3 Tests for Fixed Effects are in Table 2. It can be noted that the intercept is again not highly significant. Similarly, for a zero value of $P_{mon}$, it would be expected that no change in the *PT* value would be observed, therefore no intercept should theoretically exist. As was the case in the *POC* models, a the model was run again fixing a zero intercept and the resulting model is

$$\Delta PT = 0.1975 P_{mon} \quad\quad\quad (5)$$

The statistical significance for the effects in Eq 5 for fixed effects are listed in Table 2. The results of the Type 3 Tests for Fixed Effects can be found in Table 2. It can be noted that the $P_{mon}$ parameter remained highly significant.

Again, the zero-intercept model on average performed better than the original model in a comparison between the clinical data and the model predictions. Therefore, we use the zero-intercept model for ΔPT found in Eq 5 as the *PT* prediction model hereafter. The average difference between the recorded value and the model prediction over the 974 observations for ΔPT is less than 0.01 PSI with a standard deviation of 2.28.

## Discussion

A primary finding of this study is a model to predict general trends in the chest wall behavior under loading for the correction of the pectus carinatum deformity. Depending on the initial severity of the deformity, the treatment pressures prescribed, and the compliance of the patient

in wearing the brace, Eq 3 can be used to predict the reduction in the deformity as a function of time. It can also provide an estimate of the expected *POC* value at each follow-up visit to inform future prescribed treatment pressures and estimate patient compliance. This can be demonstrated through Fig 2. In Fig 2(A), three cases with the same initial *POC* value are shown with varying *PT* values. This can predict, assuming the same *PT* through the entire treatment, how long the treatment period would be to obtain a zero *POC*.

This model will enable an estimate of the time to full correction which may motivate patient compliance. While not yet implemented clinically, this model could be beneficial in informing patients as to the reasoning and effects of a prescribed treatment pressure. Fig 2(B) shows the effect on the correction time of varying hours of daily wear for the same initial *POC* and *PT*. This could be especially beneficial in motivating patients who struggle with brace compliance, because they can see the impact of their compliance in the time it takes to correct the deformity. This could be also used to inform conversations between patient and provider on surgical interventions. Investigation of the fail rate of bracing and surgical interventions was not assessed in this work. Incorporation of failure rates and looking at the prediction of failure with statistical model predictions could be a point of future work and could additionally motivate patient compliance and inform discussions on surgical options.

In the case of regular follow-up visits typical to pectus carinatum treatments, the model can be used in a piecewise model of the entire treatment plan, as demonstrated in an example case in Fig 3. The model is updated with a new clinical value at each follow up, which it will then be possible to adjust treatment plans and update the prediction for treatment duration.

A second finding from this study is a quantifiable change in chest wall characteristics while the chest wall experiences a constant displacement. From Eq 5 we can gain an estimate of the changes in chest wall characteristics due to sustained loading. As the brace does not change dimensions between visits, the most likely causes of the reduced pressure between when the brace was fitted at a given *PT* and the recorded value at a subsequent follow-up would be either a change in the patient's chest wall stiffness, or cellular remodeling of the deformity such that the deformity no longer presses against the brace with the same force. Either of these options, or a combination of them, could result in this change of pressure.

The models presented here can inform designs of future treatment devices for PC and pectus excavatum. The models presented here can also inform treatment pressures and the time between visits needed to see significant remodeling to assist in treatment plans. For example, in Fig 4, for a PT of 1 PSI and the brace being worn for 12 hours a day, the model predicts the brace will reduce the pressure it exerts on the deformity until just over 10 months, at which point the model predicts the body has remodeled such that the brace is no longer exerting pressure on the deformity. This information could be used to plan the intervals between follow-up visits to maintain appropriate pressures on the deformity.

For both models, no data was obtained for less than 18 days since the initial fitting, therefore these models are limited to predictions that exceed 18 days of patient brace-wearing compliance at a non-zero *PT*. For intermediate, or follow-up to follow-up, visits, the smallest recorded interval is 10 days. To avoid extrapolation errors, the models should not be used to predict treatment results with visits more frequent than that time interval.

Limitations of this study include the patient-reported value of wear time. To reduce costs of an instrumented brace for a large treatment groups, this measure provides an estimate of brace wear times which is affected by each patient's accuracy in recording the times worn. This leads to inherent error due to self-reported data. Further validation and improvement of the model through future work with instrumented braces could help refine the models. Upwards of 51 patients had yet to complete their treatment or discontinue follow-up visits. Additional data from these could further refine the models but as the models were developed based on visit-to-

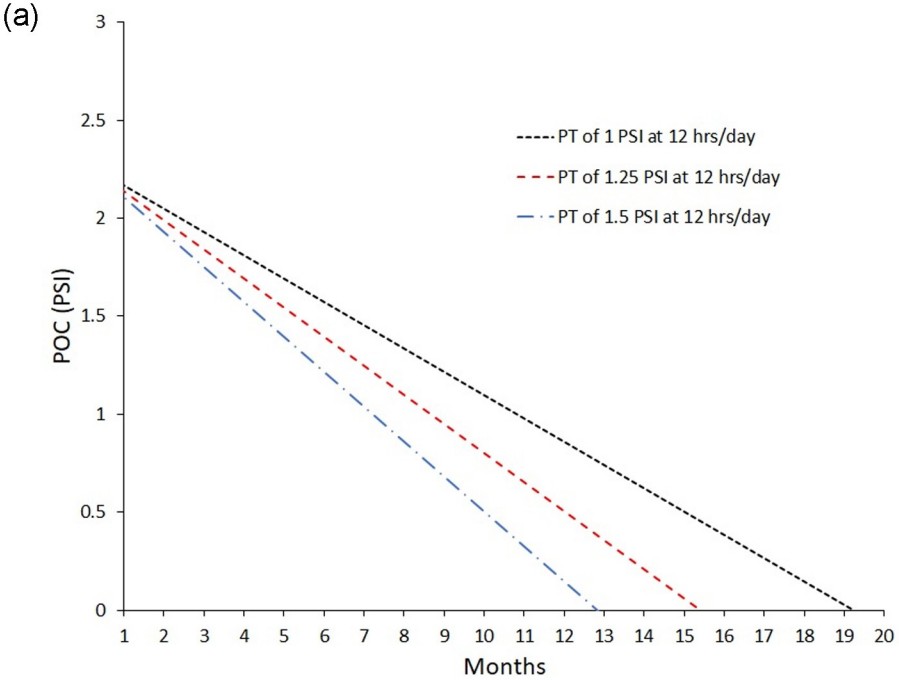

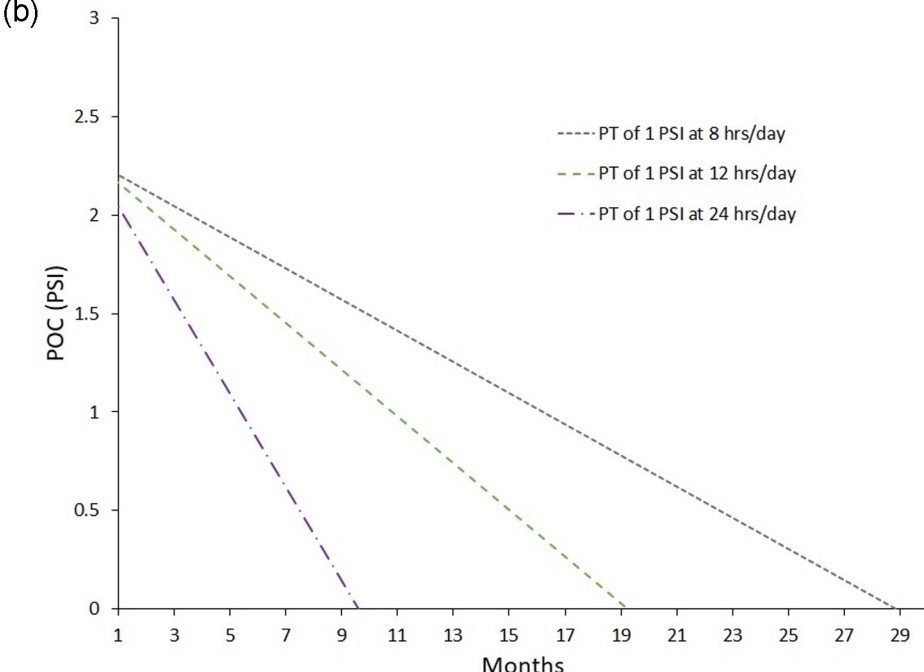

**Fig 2. Demonstrations of an example use of the POC model (Eq 3).** The predicted time to obtain a zero *POC*, assuming the same *PT* through the entire treatment, can be seen by where the lines intersect the horizontal axis. The prediction starts at 1 month as no first follow-up visit was less than 18 days from the initial fitting. (A) Three cases with the same initial *POC* value are shown with varying *PT* values. (B) Three cases demonstrating the effect of varying hours of daily wear for the same initial *POC* and *PT* on the correction time.

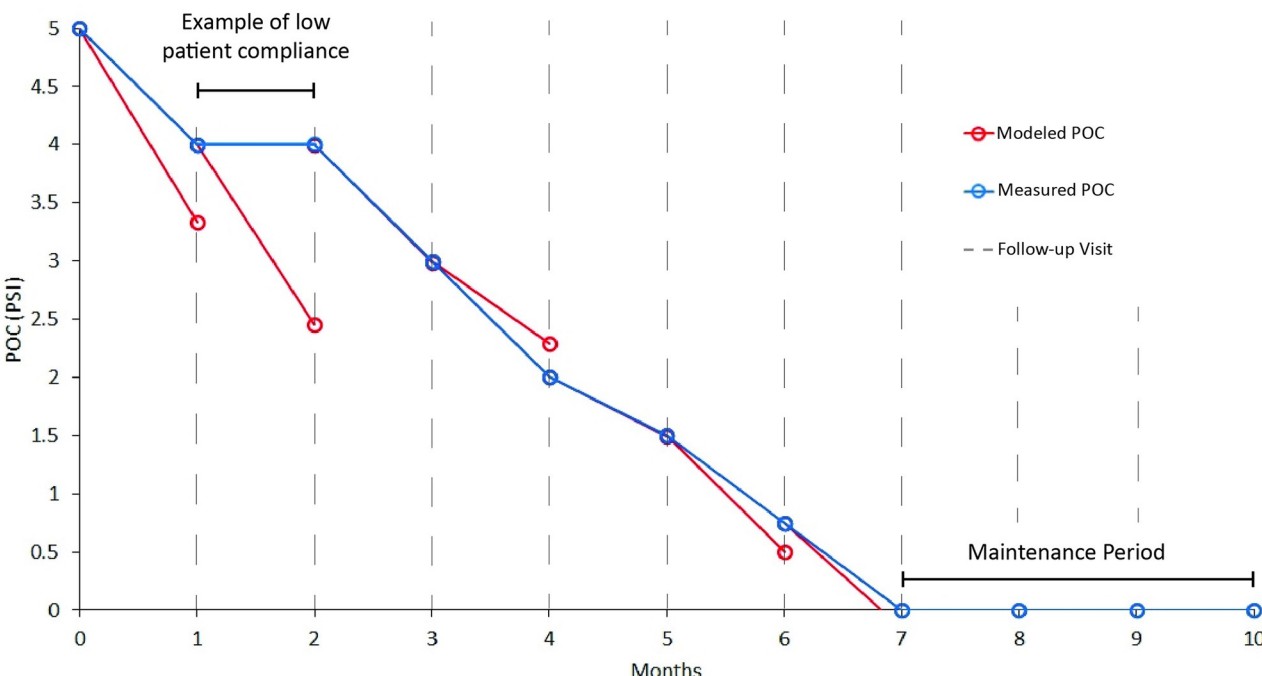

**Fig 3. Example case of using the model in a piecewise treatment plan.** The models will predict POC values which could be used to determine ideal times between follow-up visits and identify times of low patient compliance. This demonstration assumes 1 month between follow up visit. The model prediction is shown in red with fictitious patient data shown in blue. The model is updated at each follow-up visit which can then be used to alter treatment plans and adjust the predicted treatment duration.

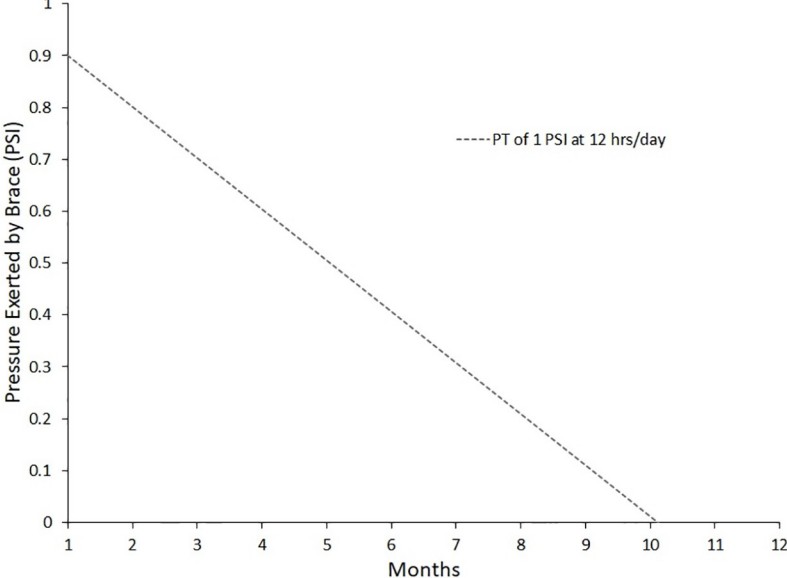

**Fig 4. An example case of using the $\Delta PT$ model found in Eq 5 at predicting the pressure exerted by the brace on the deformity as a function of time.** The example case shown has a PT of 1 PSI at 12 hrs/day of wear.

visit changes and trends, additional visits from these subjects should have little effect on the models as a whole. The models presented can be used to understand general trends across a large patient population. The data in this study did not include the factors of age and gender or other demographics which may be of interest in future studies.

The statistical models have quantified general trends in PC patients, including changes in the deformity and the changes to the chest wall under prolonged brace wear. These resulting models have the potential to assist clinical staff in treatment plan development and in motivating patient compliance. They may also be helpful in informing the design of medical devices for future treatment systems.

## Acknowledgments

The authors would like to acknowledge the work and support of the Cincinnati Children's Hospital Medical Center Chest Wall Center in data collection.

## Author Contributions

**Conceptualization:** Brandon Sargent, Victor Garcia, Larry Howell.

**Data curation:** Brandon Sargent, Katie Varela, Emily McKenna, Christina Bates.

**Formal analysis:** Brandon Sargent, Dennis Eggett.

**Funding acquisition:** Victor Garcia, Larry Howell.

**Investigation:** Brandon Sargent, Rebeccah Brown, Victor Garcia, Larry Howell.

**Methodology:** Brandon Sargent, Dennis Eggett, Victor Garcia, Larry Howell.

**Project administration:** Brandon Sargent, Rebeccah Brown, Victor Garcia, Larry Howell.

**Resources:** Emily McKenna, Christina Bates, Rebeccah Brown, Victor Garcia, Larry Howell.

**Software:** Dennis Eggett.

**Supervision:** Rebeccah Brown, Victor Garcia, Larry Howell.

**Writing – original draft:** Brandon Sargent, Katie Varela.

**Writing – review & editing:** Brandon Sargent, Katie Varela, Dennis Eggett, Victor Garcia, Larry Howell.

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
