## [Decision Letter · Decision Letter 0]

7 Jun 2023

PONE-D-23-10447Modeling of the chest wall response to prolonged bracing in pectus carinatumPLOS ONE

Dear Dr. Sargent,

Thank you for submitting your manuscript to PLOS ONE. After careful consideration, we feel that it has merit but does not fully meet PLOS ONE’s publication criteria as it currently stands. Therefore, we invite you to submit a revised version of the manuscript that addresses the points raised during the review process.

Please consider all comments

We look forward to receiving your revised manuscript.

Kind regards,

Ahmed Mancy Mosa, Ph.D.

Academic Editor

PLOS ONE

Journal Requirements:

4. Please include a copy of Table 5 which you refer to in your text on page 5.

5. We note you have included a table to which you do not refer in the text of your manuscript. Please ensure that you refer to Table 2 in your text; if accepted, production will need this reference to link the reader to the Table.

Reviewers' comments:

Reviewer's Responses to Questions

**Comments to the Author**

1. Is the manuscript technically sound, and do the data support the conclusions?

Reviewer #1: Yes

Reviewer #2: Yes

2. Has the statistical analysis been performed appropriately and rigorously? 

Reviewer #1: Yes

Reviewer #2: Yes

3. Have the authors made all data underlying the findings in their manuscript fully available?

Reviewer #1: Yes

Reviewer #2: Yes

4. Is the manuscript presented in an intelligible fashion and written in standard English?

Reviewer #1: Yes

Reviewer #2: Yes

5. Review Comments to the Author

Reviewer #1: The manuscript is interesting and good designed and written

Introduction was good written

Materials and methods were good written

Results were good described

Discussion was good written and including the conclusion of the study

Reviewer #2: The authors presented a beautifully written manuscript on creating a model of chest wall response to prolonged bracing in pectus carinatum.

As the outcomes of bracing for pectus carinatum today is normally determined via clinical outcomes by the patient and provider, have the authors seen that their modeling method leads to quicker improvement of clinical outcomes secondary to patient compliance with bracing? What have the authors seen as the fail rate of their method that requires surgical intervention and at what time point did this occur? How could providers insure adequate patient compliance and logging of bracing hours?

This method helps predict bracing time but it could also be used to help determine operative vs non-operative treatment of pectus carinatum aka if it takes x number of bracing months to achieve good clinical response is that worthwhile from the patients' standpoint vs. proceeding immediately to surgical intervention. Congratulations to the authors on their hard work.

6. PLOS authors have the option to publish the peer review history of their article (what does this mean?). If published, this will include your full peer review and any attached files.

Reviewer #1: No

Reviewer #2: No

---

## [Author Response · Author response to Decision Letter 0]

14 Jun 2023

- Response: This has been reviewed and author contributions (previously missing) has been added. 

- Response: This is still the case as we are in the process of putting the data in an open-access institutional repository. The access information for the data will be provided when available. 

- Response: This has been added to the first paragraph of the materials and methods section. 

4. Please include a copy of Table 5 which you refer to in your text on page 5.

- Response: This was a typographical error and table 5 should be table 2. This change has been made.

5. We note you have included a table to which you do not refer in the text of your manuscript. Please ensure that you refer to Table 2 in your text; if accepted, production will need this reference to link the reader to the Table.

- Response: See response to comment 4 above. 

- Response: The reference list was reviewed and no changes were made. 

Reviewer Comments to the Author

Reviewer #1: The manuscript is interesting and good designed and written

Introduction was good written

Materials and methods were good written

Results were good described

Discussion was good written and including the conclusion of the study

- Response: Thank you for you support of this work!

Reviewer #2: The authors presented a beautifully written manuscript on creating a model of chest wall response to prolonged bracing in pectus carinatum.

As the outcomes of bracing for pectus carinatum today is normally determined via clinical outcomes by the patient and provider, have the authors seen that their modeling method leads to quicker improvement of clinical outcomes secondary to patient compliance with bracing? What have the authors seen as the fail rate of their method that requires surgical intervention and at what time point did this occur? How could providers insure adequate patient compliance and logging of bracing hours?

This method helps predict bracing time but it could also be used to help determine operative vs non-operative treatment of pectus carinatum aka if it takes x number of bracing months to achieve good clinical response is that worthwhile from the patients' standpoint vs. proceeding immediately to surgical intervention. Congratulations to the authors on their hard work.

- Response: Thank you for your comments and suggestions. The following discussion on operative and non-operative approaches discussion for patients and providers has been added as well as discussion on future work in comparing the models to improved clinical outputs, fail rates, and patient compliance. These discussions were added to the “Discussion” section.

---

## [Decision Letter · Decision Letter 1]

7 Jul 2023

Modeling of the chest wall response to prolonged bracing in pectus carinatum

PONE-D-23-10447R1

Dear Dr. Sergent,

We’re pleased to inform you that your manuscript has been judged scientifically suitable for publication and will be formally accepted for publication once it meets all outstanding technical requirements.

Kind regards,

Ahmed Mancy Mosa, Ph.D.

Academic Editor

PLOS ONE

Additional Editor Comments (optional):

Reviewers' comments:

Reviewer's Responses to Questions

**Comments to the Author**

1. If the authors have adequately addressed your comments raised in a previous round of review and you feel that this manuscript is now acceptable for publication, you may indicate that here to bypass the “Comments to the Author” section, enter your conflict of interest statement in the “Confidential to Editor” section, and submit your "Accept" recommendation.

Reviewer #2: All comments have been addressed

2. Is the manuscript technically sound, and do the data support the conclusions?

Reviewer #2: Yes

3. Has the statistical analysis been performed appropriately and rigorously? 

Reviewer #2: Yes

4. Have the authors made all data underlying the findings in their manuscript fully available?

Reviewer #2: Yes

5. Is the manuscript presented in an intelligible fashion and written in standard English?

Reviewer #2: Yes

6. Review Comments to the Author

Reviewer #2: The authors have addressed all my comments. This study will be a great contribution to the literature and I congratulate the authors on their work.

7. PLOS authors have the option to publish the peer review history of their article (what does this mean?). If published, this will include your full peer review and any attached files.

Reviewer #2: No

---

## [Editor Report · Acceptance letter]

1 Aug 2023

PONE-D-23-10447R1 

Modeling of the chest wall response to prolonged bracing in pectus carinatum 

Dear Dr. Sargent:

I'm pleased to inform you that your manuscript has been deemed suitable for publication in PLOS ONE. Congratulations! Your manuscript is now with our production department. 

Kind regards, 

on behalf of

Dr. Ahmed Mancy Mosa 

Academic Editor

PLOS ONE